# Diagnostic test accuracy of artificial intelligence in screening for referable diabetic retinopathy in real-world settings: A systematic review and meta-analysis

Holijah Uy[1]*, Christopher Fielding[2], Ameer Hohlfeld[3], Eleanor Ochodo[4,5], Abraham Opare[1], Elton Mukonda[2], Deon Minnies[1‡], Mark E. Engel[3,6‡]*

1 Community Eye Health Institute, Faculty of Health Sciences, University of Cape Town, Cape Town, South Africa, 2 Division of Epidemiology and Biostatistics, School of Public Health, Faculty of Health Sciences, University of Cape Town, Cape Town, South Africa, 3 South African Medical Research Council, Cape Town, South Africa, 4 Centre for Global Health Research, Kenya Medical Research Institute, Nairobi, Kenya, 5 Centre for Evidence-Based Health Care, Department of Global Health, Faculty of Medicine and Health Sciences, Stellenbosch University, Stellenbosch, South Africa, 6 Department of Medicine, University of Cape Town, Cape Town, South Africa

‡ DM and MEE are joint senior authors on this work.
* mark.engel@uct.ac.za (MEE); uyxhol001@myuct.ac.za (HU)

**Data Availability Statement:** The data incorporated into the systematic review are from

## Abstract

Retrospective studies on artificial intelligence (AI) in screening for diabetic retinopathy (DR) have shown promising results in addressing the mismatch between the capacity to implement DR screening and increasing DR incidence. This review sought to evaluate the diagnostic test accuracy (DTA) of AI in screening for referable diabetic retinopathy (RDR) in real-world settings. We searched CENTRAL, PubMed, CINAHL, Scopus, and Web of Science on 9 February 2023. We included prospective DTA studies assessing AI against trained human graders (HGs) in screening for RDR in patients with diabetes. Two reviewers independently extracted data and assessed methodological quality against QUADAS-2 criteria. We used the hierarchical summary receiver operating characteristics (HSROC) model to pool estimates of sensitivity and specificity and, forest plots and SROC plots to visually examine heterogeneity in accuracy estimates. From our initial search results of 3899 studies, we included 15 studies comprising 17 datasets. Meta-analyses revealed a sensitivity of 95.33% (95%CI: 90.60–100%) and specificity of 92.01% (95%CI: 87.61–96.42%) for patient-level analysis (10 datasets, N = 45,785) while, for the eye-level analysis, sensitivity was 91.24% (95%CI: 79.15–100%) and specificity, 93.90% (95%CI: 90.63–97.16%) (7 datasets, N = 15,390). Subgroup analyses did not provide variations in the diagnostic accuracy of country classification and DR classification criteria. However, a moderate increase was observed in diagnostic accuracy in the primary-level healthcare settings: sensitivity of 99.35% (95%CI: 96.85–100%), specificity of 93.72% (95%CI: 88.83–98.61%) and, a minimal decrease in the tertiary-level healthcare settings: sensitivity of 94.71% (95%CI: 89.00–100%), specificity of 90.88% (95%CI: 83.22–98.53%). Sensitivity analyses did not show any variations in studies that included diabetic macular edema in the RDR definition, nor studies with ≥3 HGs. This review provides evidence, for the first time from prospective studies, for

published data, and are thus freely available as per the references supplied.

**Funding:** The authors received no specific funding for this work.

**Competing interests:** The authors have declared that no competing interests exist.

the effectiveness of AI in screening for RDR in real-world settings. The results may serve to strengthen existing guidelines to improve current practices.

## Introduction

Diabetic retinopathy (DR) is the most common and specific complication of diabetes mellitus in the working age group [1]. In 2020, the number of adults with DR was estimated to be 103.12 million, which is expected to be 129.84 million by 2030 and 160.50 million by 2045 [1]. Along with an increasing incidence of DR, the number of people with vision impairment and blindness also increases. Without early intervention, the incidence of blindness due to DR will continue to rise as the number of people getting diabetes increases. Thus, DR has become a global public health concern, compelling researchers and health practitioners to continuously develop strategies to prevent and treat DR.

Diabetic retinopathy can be asymptomatic for years, even at advanced stages [2]. Thus, early-stage detection of DR is crucial to provide timely treatment and management. For that reason, DR screening programmes are being implemented in public health settings through population-based or opportunistic screening. Diabetic retinopathy screening aims to distinguish between patients who need a referral, termed referable DR (RDR), for ophthalmological intervention from those who can continue annual routine eye care services [3]. Referable DR can be classified as moderate nonproliferative DR (NPDR) or worse and/or diabetic macular edema (DME). Those with RDR must be referred within three months to one year, depending on the resource settings [4]. Conventional methods for DR screening include the use of direct ophthalmoscopy, slit lamp biomicroscopy, mydriatic and nonmydriatic retinal photography, and retinal video recording. Examiners performing these methods include trained non-medical degree personnel, ophthalmic nurses, optometrists, general practitioners, ophthalmologists, and retina specialists; however, Hazin et al. argued that when interpreting retinal images, the inclusion of ophthalmologists should be ensured [5].

Currently, local and international programmes combatting DR are facing a significant crisis due to the increasing prevalence of diabetes. This influx has outpaced the development of healthcare services and screening programmes for preventing DR [6]. According to a systematic review by Piyasena et al. [7], aside from the high cost of services and lack of infrastructure for retinal imaging and training programmes, one of the major barriers to DR screening is the lack of skilled human resources, especially in the lower- and middle-income countries. The advent of digital ophthalmic solutions in recent years, such as retinal screening through tele-ophthalmology, has attracted great interest from both the medical and public health communities, where these barriers exist. Teleophthalmology was meant to target most of these barriers [8]. However, teleophthalmology alone is not enough to address these barriers, especially with regards to waiting times and human resources.

Artificial intelligence has shown to be a promising solution to these challenges by functioning in an autonomous mode. Through deep learning algorithms, AI can be used to detect the presence and severity of DR in real-time. However, it is crucial that these tools should have high diagnostic accuracy and good performance before being implemented in various healthcare settings. The UK National Institute for Clinical Excellence (NICE) Guidelines stated that DR screening programmes should use screening tools with a sensitivity of $\geq$80%, specificity of $\geq$95%, and a technical failure rate of $\leq$5% [9]. Meanwhile, the St Vicent Declaration of 2005

suggested that systematic DR screening programmes should aim for a sensitivity of ≥80% and a specificity of ≥90% with an acceptable coverage of ≥ 80% [10].

In recent years, retrospective validation studies have shown AI to have high diagnostic accuracy in detecting DR [11, 12]; that is, AI is equally good or even better than human graders (HGs). Studies done in real-world settings using prospective data collection have also demonstrated robust performance [13]; however, these studies are fewer than those done in a retrospective manner, and the true utility of AI systems in DR screening will only be better understood through prospective studies, as performance is likely to be affected when dealing with real-world data that is different from the data used for algorithm training [14, 15]. Moreover, prospective studies, with pre-established protocols, allow them to be more robust and generalisable, and exhibit the true impact on system usability in real-world settings.

Therefore, we conducted a systematic review and meta-analysis of studies with prospective data collection in assessing the diagnostic accuracy of AI compared with trained HGs in screening for RDR in real-world settings. The findings of this review may offer evidence-based recommendations for integrating AI solutions to screen for RDR, especially in resource-challenged environments.

## Methods

### Reporting, protocol, and registration

We drafted this review in accordance with the Preferred Reporting Items for Systematic Review and Meta-analysis of Diagnostic Test Accuracy Studies (PRISMA-DTA) guidelines [16]. The study protocol was registered with the International Prospective Register of Systematic Reviews (PROSPERO) under CRD42023392297. An ethics waiver was granted by the University of Cape Town Human Research Ethics Committee.

### Databases and search strategies

We searched the following electronic databases: Cochrane Central Register of Controlled Trials (CENTRAL), Medical Literature Analysis and Retrieval System Online (MEDLINE) via PubMed, Cumulative Index to Nursing and Allied Health Literature (CINAHL), Scopus, and Web of Science on 9 February 2023 with no language restrictions (S1 Table). We also hand-searched the reference lists of relevant primary studies, systematic reviews, and the following journals: British Journal of Ophthalmology, American Journal of Ophthalmology, Ophthalmology and Retina, JAMA Ophthalmology, and Investigative Ophthalmology and Visual Science; and all these studies were already included or accounted for from the electronic database search, thus confirming the search strategy to be comprehensive.

### Eligibility criteria

**Type of studies.**   We included randomised control trials (RCT) and observational analytical studies evaluating the DTA of AI in DR screening. We excluded studies based on retrospective validation of existing images (i.e., medical records, available data sets). We excluded review articles, editorials, case series, case reports, and qualitative studies.

**Type of participants.**   We included participants with clinically diagnosed type 1 or type 2 diabetes with unknown DR status, regardless of age, sex, race/ethnicity, and geographical location. We excluded studies that enrolled participants with unconfirmed diabetes to avoid misclassifying participants, which may result in biased estimates of the association between diabetes and diabetic retinopathy.

**Setting.** We only included studies conducted in real-world settings, thus excluding those done for theoretical algorithm training and validation alone.

**Index test.** We included interventions using AI for prospective screening of fundus images that could detect RDR or its equivalent.

**Reference standard.** The reference standard was manual grading for DR by trained HGs who analysed the same fundus images read by the AI. We excluded reference standards that did not use the same DR classification criteria used by the AI during its software training to grade DR.

**Target condition.** We included studies that screened for RDR as defined by the authors of the primary studies. We included studies with RDR equivalence, i.e. more than mild DR, clinically significant DR, etc. We did not include patients or eyes with no RDR, and ungradable or inconclusive fundus images in the pooling of diagnostic accuracy outcomes. Including ungradable or inconclusive images may result in inaccuracy in assessing the AI system's performance, making it challenging to draw meaningful conclusions.

**Outcomes.** We included studies reporting on, or containing the data necessary to extract information on the proportions of true positives (TP), false positives (FP), true negatives (TN), and false negatives (FN). Efforts were made to contact corresponding authors to retrieve data which were unclear or unavailable in the paper or supplementary materials.

## Study selection

We used Rayyan software to manage the retrieved studies. Review authors (HU, CF) independently screened the titles and abstracts and classified them as (a) included, (b) maybe, and (c) excluded. Full-text articles of those 'included' and 'maybe' were obtained and independently assessed by the same authors against the eligibility criteria. Studies were then classified as (a) included, (b) excluded, and (c) awaiting authors' responses. Any disagreements were resolved between the two reviewers through discussion or consulting a third review author (AH). We emailed the corresponding authors of studies included as 'awaiting authors' responses' at least three times with intervals of at least two weeks. If there were no responses from the authors, studies were classified under 'no author's response'.

## Data extraction and management

We developed a data extraction form and divided it into two parts: (a) Study characteristics (relating to study designs, AI, and reference standards) and (b) Study outcomes: TP, FP, FN, TN. Two review authors extracted the study characteristics and study outcomes.

## Risk of bias and applicability

The risk of bias and applicability on the (a) patient selection, (b) index test, (c) reference standard, and (d) flow and timing of the included studies were independently assessed by two review authors (HU, CF) using the Quality Assessment of Diagnostic Accuracy Studies (QUADAS)-2 tool [17]. We tailored, piloted, and refined our QUADAS-2 tool based on our review. Any disagreements were resolved between the two authors through discussion or consulting a senior author (ME).

## Data synthesis and analysis

**Quantitative data analysis and synthesis.** We calculated each included study's sensitivity and specificity. We initially planned to analyse data only at the patient level; however, some studies reported only diagnostic accuracy on eye level (or image level), and some patient-level

data cannot be extracted. Therefore, we considered looking into both of these levels for analysis. Heterogeneity was explored using visual inspection of forest plots and hierarchical summary receiver operating characteristics (HSROC) plots. All analyses performed and plots generated were done using Review Manager (RevMan) 5.4 and SAS Studio.

**Subgroup analysis.** We performed subgroup analyses on the following covariates identified a priori: level of economic development (World Bank country classification), level of the healthcare setting, and DR classification criteria. We did not include the modes of AI as previously planned since all AI modes of the included studies were automated.

**Sensitivity analysis.** We initially planned to explore the effect of excluding studies with a high risk of bias. However, after excluding studies with a high risk of bias, all studies were left with an unclear risk. Nevertheless, we performed sensitivity analyses to investigate the exclusion of studies that did not include DME in the RDR definition; although we have stated that the definition of RDR will be according to how the authors of the primary studies defined it, many references still included DME as part of RDR definition, and the International Council of Ophthalmology (ICO) guidelines states that patients with DME should be referred. We also investigated the exclusion of studies with ≤2 HGs as the ground truth for reference standard because this might incur bias if intergrader disagreements arise without having a third HG to arbitrate. According to Cardoso et al., ground truth means "data and/or method related to more consensus or reliable values/aspects that can be used as references" [18]. In our review, it refers to the final grading or assessment of fundus images by all HGs, which serves as the reference standard or the most reliable evaluation of the presence and severity of DR.

## Results

### Results of the search

We were able to identify a total of 3899 articles through searching of various databases. After deduplication, 2742 studies were screened by title/abstract, of which 2654 were excluded. The remaining 88 studies were screened for full-text assessment against the review's eligibility criteria. Of these, 70 studies were excluded, three were classified under 'no author's response', and finally, 15 were included for the quantitative synthesis (Fig 1).

**Included studies.** Please see Table 1a and 1b for the characteristics of included studies. Fifteen studies comprising 17 datasets were deemed eligible for this review, of which ten measured diagnostic accuracies at the patient level (45 785 patients) and seven at the eye level (15 390 eyes). We deemed all studies to be cross-sectional with prospective data collection; however, in our table of included studies, we presented the study designs according to how they were reported. Seven studies were done in China, five in India, and three in Australia. Seven studies were done at tertiary-level healthcare settings, six were done at the primary level, while the remaining two were done at both levels; no studies were done at the secondary level. For the target condition (RDR), six studies defined it as moderate NPDR or worse and/or DME, and nine did not include DME as part of the definition. Thirteen studies used ICDR or its equivalence as DR classification criteria, and two used the NHS DES criteria. For the reference standard, 12 studies have ≥3 HGs as the ground truth, and three studies have at most two HGs. Four studies developed their own AI models, and 11 used commercially available models. All studies used Inception with varying versions as their architecture. All AI software in the studies were fine-tuned with training data sets containing 25 297 to 207 228 fundus images. All studies used nonmydriatic cameras to capture fundus images, of which three still performed mydriasis on their patients using tropicamide eye drops, and one did mydriasis on a conditional protocol. Eight studies captured only one fundus field per eye (mostly

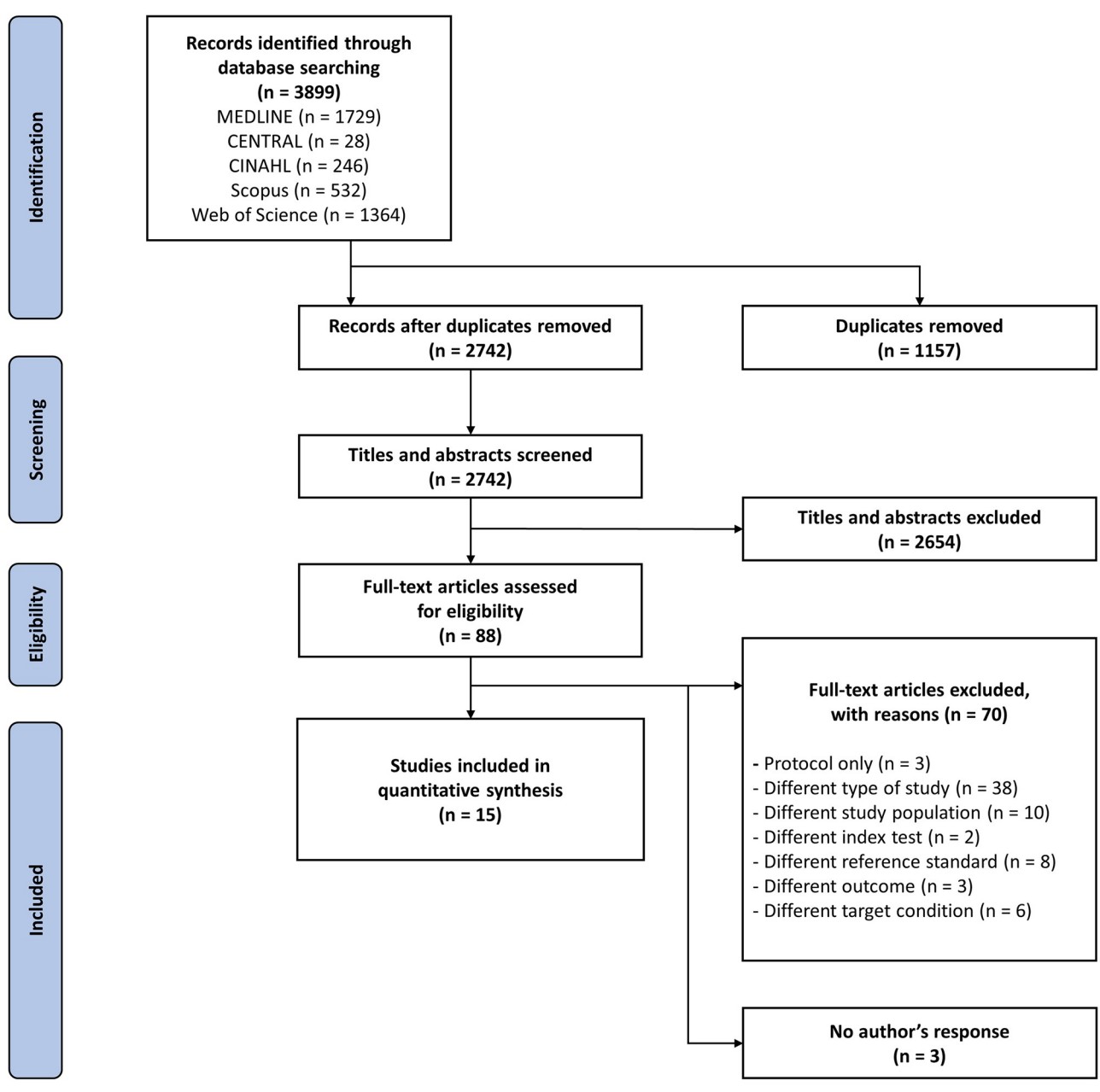

**Fig 1. PRISMA flow diagram of the study search and selection.**

macula-centred), and seven studies captured more than one fundus field. All studies used a fundus camera with a narrow field of vision (45˚-50˚).

**Excluded studies.** From the 88 full-text articles assessed for eligibility, we excluded 70 studies and classified three studies under 'no author's response'.

## Methodological quality of included studies

A summary of methodological quality assessment is presented in Figs 2 and 3.

**Patient selection.** In the patient selection domain, 12 of the 15 studies were deemed to have an unclear risk of bias in the sampling method. Most of the studies did not specify how

**Table 1.** a. Key characteristics of the study design, population, target condition, and reference standard of included studies. b. Key characteristics of the index tests of included studies.

| Study | Study Settings | | | Patient Characteristics | | | | Target Condition | | Reference Standard/ Ground Truth (№) | |
|---|---|---|---|---|---|---|---|---|---|---|---|
| | Study Design [a] | Country (WBC) | Setting (№) | Healthcare Setting | Age, mean (SD), years | Type of Diabetes | Sample Size [b] (Patients/ Eyes) | Definition of RDR/ Equivalence | Criteria Used | If without disagreement | If with disagreement |
| Dong 2022 [19] | Cross-sectional | China (U-MIC) | Community healthcare centres (3) | Primary | 52.09 (±11.51) | T1D, T2D | Eyes: 848 | Moderate NPDR or worse (DME not included) | ICDR | Gradings made by the ophthalmologists (2) | Gradings made by a senior retinal specialist (1) |
| Gulshan 2019 [20] | Prospective observational | India (L-MIC) | Eye care centre (Aravind Eye Hospital only) | Tertiary | 56.60 (±9.00) | T1D, T2D | Eyes: 1905 [c] | Moderate NPDR or worse (DME not included) | ICDR | Gradings made by retinal specialists (3) | any disagreements discussed until a full consensus was achieved |
| Hao 2022 [21] | Prospective clinical trial | China (U-MIC) | Local community hospital | Primary | 63.03 (±8.72) | T1D, T2D | Eyes: 6854 | Moderate NPDR or worse (DME not included) | ICDR | Gradings made by the ophthalmologists (2) | Gradings made by a senior ophthalmologist (1) |
| He 2020 [22] | Cross-sectional [d] | China (U-MIC) | Community hospital clinic | Primary | 68.46 (±7.20) | T1D, T2D | Patients: 889 | Moderate NPDR or worse and/or DME | ICDR | Gradings made by the retina specialists (2) | Gradings made by a third retinal specialist (1) |
| Jain 2021 [23] | Cross-sectional | India (L-MIC) | Municipal dispensaries (47) | Primary | 54.90 (±10.43) | T1D, T2D | Patients: 1370 Eyes: 2626 | Moderate NPDR or worse (DME not included) | ICDR | Gradings made by the retina specialists (2) | Gradings made by a third retinal specialist (1) |
| Kanagas-ingam 2018 [24] | Cross-sectional [d] | Australia (HIC) | Primary care clinic | Primary | 55.00 (±17.00) | T1D, T2D | Patients: 193 | Moderate NPDR or worse (DME not included) | ICDR | Grading made by an ophthalmologist (1) alone | |
| Keel 2018 [25] | Prospective observational | Australia (HIC) | Urban endocrinology outpatient clinics (2) | Tertiary | 44.26 (±16.56) | T1D, T2D | Patients: 93 | Moderate NPDR or worse and/or DME | NHS DES | Grading made by the centralised retinal grading centre | |
| Li 2021 [26] | Prospective observational | China (U-MIC) | General hospital | Tertiary | 50.00 (±12.00) | T1D, T2D | Eyes: 1674 [c] | Moderate NPDR or worse (DME not included) | ICDR | Grading made by a retina specialist (1) alone | |
| Natarajan 2019 [27] | Prospective, cross-sectional | India (L-MIC) | Municipal dispensaries | Primary | 53.10 (±10.30) | T1D, T2D | Patients: 214 Eyes: 394 | Moderate NPDR and worse, with or without DME | ICDR | Grading made by the ophthalmology resident (1) and retina specialist (1) | Gradings made by the same retina specialist |
| Rajalakshmi 2018 [28] | Cross-sectional [d] | India (L-MIC) | Diabetes centre | Tertiary | NR | T2D | Patients: 296 | Moderate NPDR or worse and/or DME | ICDR | Grading made by the retina specialists (2) | Gradings made by a third retinal specialist (1) |
| Scheetz 2021 [29] | Prospective observational | Australia (HIC) | Endocrinology outpatient clinics (2) and Aboriginal medical services clinics (3) | Primary and Tertiary | 54.25 (±20.16) [e] | T1D, T2D | Patients: 203 | Moderate NPDR or worse and/or DME | NHS DES | Gradings made by NHS-certified graders (2) | Gradings made by retinal specialists (2) |
| Sosale 2020 [30] | Prospective, cross-sectional | India (L-MIC) | Diabetes centre | Tertiary | NR | T1D, T2D | Patients: 900 | Moderate NPDR or worse and/or DME | ICDR | The majority diagnosis of the retina specialists (5) | |
| Yang 2022 [31] | Observational, prospective, multicentre, gold standard-controlled | China (U-MIC) | Hospital and ophthalmic centres (3) | Tertiary | 60.44 (±10.19) [e] | T1D, T2D | Patients: 962 | Stage II or worse DR (DME not included) | COS [f] | Gradings made by ZIRC graders (2) | Gradings made by a third senior ZIRC grader (1) |
| Zhang 2020 [32] | Prospective observational | China (U-MIC) | Diabetes centres (155) | Primary and Tertiary | 54.29 (±11.60) | T1D, T2D | Patients: 40 665 | Moderate NPDR or worse (DME not included) | ICDR | Gradings made by the ophthalmologists (2) | Gradings made by a senior ophthalmologist (1) |
| Zhang 2022 [33] | Prospective, multicentre, self-controlled clinical trial | China (U-MIC) | Hospitals (3) | Tertiary | 56.52 (±11.13) | T1D, T2D | Eyes: 1089 | Moderate NPD or worse (DME not included) | ICDR | Gradings made by the ophthalmologists (3) | Gradings made by the principal investigator ophthalmologist (1) |

| Study | Artificial Intelligence Development | | | | | | Fundus Camera Used | | |
|---|---|---|---|---|---|---|---|---|---|
| | AI Model | Architecture | Neural Network | Pre-trained | Fine-tuned | Training Dataset (№ of fundus images) | Mydriatic or Nonmydriatic Camera | № of Fundus Fields | Field of Vision |
| Dong 2022 [19] | C.ARE, Shanghai EagleVision Medical Technology Co., Ltd (Airdoc) | Inception-ResNet-v2 | CNN | Yes | Yes | Clinical settings datasets (207 228) | Nonmydriatic | 1 field (macula-centred) | 50° |
| Gulshan 2019 [20] | Own AI model | Inception-v3 | CNN | Yes | Yes | EyePACS and hospital datasets (128 175) | Nonmydriatic | 1 field (macula-centred) | 45° |
| Hao 2022 [21] | EyeWisdom (Visionary Intelligence Ltd., Beijing, China) | Inception-v3 | CNN | Yes | Yes | EyePACS and hospital datasets (25 297) | Nonmydriatic | 2 fields (macula- and optic disc-centred) | 45° |
| He 2020 [22] | Airdoc, Beijing, China | Inception-v4 | CNN | Yes | Yes | Unspecified dataset (number of fundus images NR) | Nonmydriatic | 2 fields (macula- and optic disc-centred) | 45° |
| Jain 2021 [23] | Medios AI (Remidio) | Inception-v3 and MobileNet | CNN | Yes | Yes | EyePACS, hospital and screening camps datasets (52 894) | Nonmydriatic, but patients underwent mydriasis (1% tropicamide) | 3 fields (posterior pole including macula & disc, nasal, and temporal) | 45° |

(Continued)

| Study | AI model | Algorithm | | | | Dataset (sample size) | Mydriasis | Fields | Angle |
|---|---|---|---|---|---|---|---|---|---|
| Kanagas-ingam 2018 [24] | Own AI model | Inception-v3 (customised) | CNN | Yes | Yes | Yes | DiaRetDB1, EyePACS, and Tele-eye care DR database (30 000) | Nonmydriatic | 1 field (macula-centred) | 45° |
| Keel 2018 [25] | Own AI model | Inception-v3 | CNN | NR | Yes | Yes | LabelMe dataset (58 790) | Nonmydriatic | 1 field (central nasal) | 45° |
| Li 2021 [26] | VoxelCloud, China | Inception-ResNet-v2 | CNN | Yes | Yes | Yes | EyePACS and hospital datasets (141 184) | Nonmydriatic | 1 field (macula-centred) | 45° |
| Natarajan 2019 [27] | Medios AI (Remidio) | Inception-v3 and MobileNet | CNN | Yes | Yes | Yes | EyePACS, hospital and screening camps datasets (52 894) | Nonmydriatic, but patients underwent mydriasis (1% tropicamide) | 3 fields (posterior pole including macula & disc, nasal, and temporal) | 45° |
| Rajalakshmi 2018 [28] | EyeArt v2.1 | NR | DNN | Yes | Yes | Yes | EyePACS (number of fundus images NR) | Nonmydriatic, but patients underwent mydriasis (tropicamide) | 4 fields (macula-centred, optic disc-centred, superior-temporal, and inferior-temporal quadrants of the retina) | 45° |
| Scheetz 2021 [29] | Own AI model | Inception-v3 | CNN | NR | Yes | Yes | LabelMe dataset (71 043) | Nonmydriatic | 1 field (macula-centred) | 45° |
| Sosale 2020 [30] | Medios AI (Remidio) | Inception-v3 and MobileNet | CNN | Yes | Yes | Yes | EyePACS, hospital and screening camps datasets (52 894) | Nonmydriatic | 2 fields (macula- and optic disc-centred) | 45° |
| Yang 2022 [31] | AIDRScreening v1.0 (Shenzhen SiBright CO. Ltd., China) | NR | CNN | NR | Yes | Yes | Eye institute, endocrinology department, and eye examination centre datasets (73 849) | Both; if pupil diameter was >4 mm, fundus photography was performed without mydriasis; otherwise, mydriasis was required | 2 fields (macula- and optic disc-centred) | 45° |
| Zhang 2020 [32] | VoxelCloud Retina, China | Inception-ResNet-v2 | CNN | Yes | Yes | Yes | EyePACS and hospital datasets (144 810) | Nonmydriatic | 1 field (macula-centred) | 45° |
| Zhang 2022 [33] | EyeWisdom v1 (Visionary Intelligence Ltd., Beijing, China) | Inception-v3 and ResNet-34 | CNN | Yes | Yes | Yes | Hospital and ILSVRC subset of ImageNet datasets (40 693) | Nonmydriatic | 1 field (posterior pole containing macula and optic disc) | 45° |

[a] Study design according to study authors;

[b] Sample included in the diagnostic accuracy analysis excluding ungradable images;

[c] Samples were reported in image level, but the study captured one image per eye, so considered as eye-level;

[d] Study design not reported, thus deemed by review authors as cross-sectional based on the journals;

[e] Mean was estimated from median using recommendations by Hong Kong Baptist University, Department of Mathematics [34];

[f] Criteria was matched to the equivalent definition of RDR based on the ICDR classification.

**COS**, Chinese Ophthalmic Society; **CSME**, clinically significant macular edema; **DME**, diabetic macular edema; **DR**, diabetic retinopathy; **ETDRS**, Early Treatment Diabetic Retinopathy Study; **HIC**, high-income country; **ICDR**, International Clinical Diabetic Retinopathy; **L-MIC**, lower middle-income country; **NHS DES**, National Health Service Diabetic Eye Screening; **NPDR**, nonproliferative diabetic retinopathy; **NR**, not reported; **RDR**, referable diabetic retinopathy; **SD**, standard deviation; **T1D**, Type 1 diabetes; **T2D**, Type 2 diabetes; **U-MIC**, upper middle-income country; **WBC**, World Bank classification; **ZIRC**, Zhongshan Image Reading Centre.

**CARE**, Comprehensive AI Retinal Expert; **CNN**, convolutional neural network; **DNN**, deep neural network; **ILSVRC**, ImageNet Large Scale Visual Recognition Challenge; **NR**, not reported

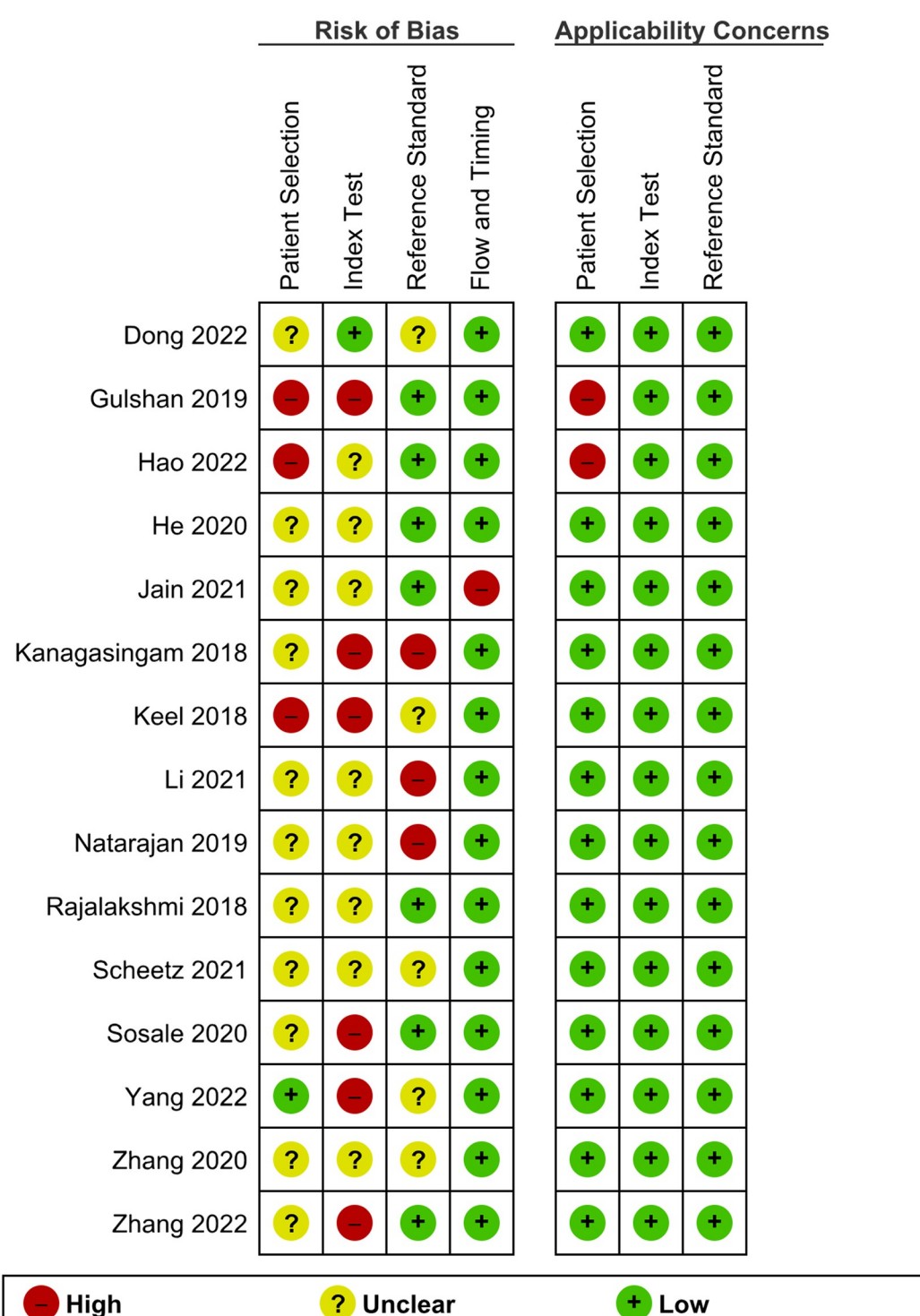

**Fig 2. Risk of bias and applicability concerns summary: Review authors' judgments about each domain for each included study using the QUADAS-2 tool.**

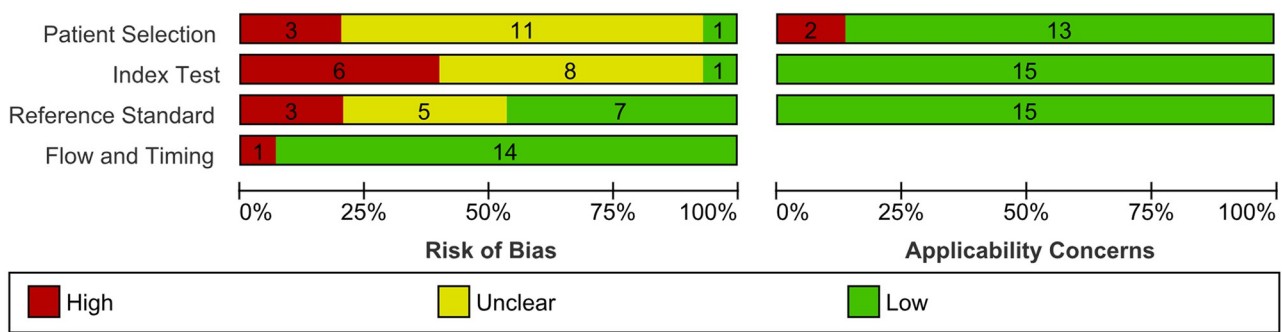

**Fig 3. Risk of bias and applicability concerns graph: Review authors' judgments about each domain presented as percentages across included studies using the QUADAS-2 tool.**

patients were enrolled except for three studies (1 consecutive, 1 random, and 1 convenience sampling method). Two studies were not able to avoid inappropriate exclusions since one study excluded patients with macular edema, and the other study excluded those who were treated with ocular injections for DME or proliferative disease; of which these conditions are part of the definition of RDR, deeming these studies with a high risk of bias and high concern on applicability. For applicability on patient selection, 13 out of 15 studies have a low concern on applicability.

**Index test.**   In the domain of index tests, we added two signalling questions deemed necessary for index tests using AI, one of which is the quality of images fed into the AI system. This is vital since images with insufficient quality (i.e., overexposed, out-of-focus, etc.) may be deemed ungradable or be misclassified. Another signalling question added was on the conflict of interest. With the advent of AI in healthcare, several AI software packages are currently being developed; thus, if study authors were affiliated with or funded by the software company in any way, studies may incur a high risk of bias.

In this domain, the main quality issue was the signalling question of whether a diagnostic threshold was prespecified or not. Only three studies reported on prespecified thresholds, with the remaining 12 studies thus considered to have an unclear risk of bias. Six studies have conflicts of interest, thus deeming them high risk. All studies have low applicability concerns for the index test.

**Reference standard.**   In the reference standard domain, three out of 15 studies were evaluated as having a high risk of bias because there were only two HGs to grade the fundus images. This may incur bias since grading images can be very subjective, and there is no one to arbitrate when a disagreement arises. Five studies have an unclear risk of bias because they did not explicitly state whether the HGs were blinded to the results of the AI grading results. All studies have low applicability concerns.

**Flow and timing.**   In the domain of flow and timing, one study was considered to be of a high risk of bias because it was not able to explain the discrepancies in patients enrolled and analysed clearly. This domain is not assessed regarding applicability concerns, as stated in the QUADAS-2 tool.

## Findings

We evaluated the accuracy of AI in screening for RDR in real-world settings according to patient-level and eye-level analysis compared with HGs. The patient-level analysis was considered the main meta-analysis since it is the number of patients with RDR who will be referred to ophthalmologists for further assessment. Out of the 15 studies reviewed, eight presented

**Table 2. Overall patient-level and eye-level meta-analysis of the accuracy of AI in detecting RDR compared with trained HGs.**

| Overall Meta-analysis | № of Studies | № of Samples | Sensitivity (95% CI) | Specificity (95% CI) |
|---|---|---|---|---|
| Patient-level | 10 | 45 785 patients | 95.33% (90.60–100) | 92.01% (87.61–96.42) |
| Eye-level | 7 | 15 390 eyes | 91.24% (79.15–100) | 93.90% (90.63–97.16) |

Data calculated using SAS Studio.

**CI**, Confidence Interval; **HG**, human grader.

diagnostic accuracy based solely on patient-level information, five showed diagnostic accuracy based solely on eye-level information, and two showed diagnostic accuracy based on both patient-level and eye-level information.

The HSROC model by Rutter and Gatsonis was used for the meta-analysis as this model accounts for the variations in the test thresholds among the AI models [35]. We performed subgroup analysis and investigated for heterogeneity using the World Bank country classification, level of the healthcare setting, and DR classification criteria.

We performed sensitivity analyses to explore the effect of excluding (1) studies that did not include DME in the RDR definition and (2) studies with a total number of ≤2 HGs as the ground truth. Table 2 shows the detailed overall patient-level and eye-level meta-analysis.

**Patient-level analysis.** Ten evaluations of AI for RDR screening were performed with data from ten studies and a total of 45 785 patients. The forest plot (Fig 4) shows minimal variation in the accuracy estimates. The HSROC plot (Fig 5) reveals good test accuracy since most study points lie in the upper left corner of the plot. Meta-analytical sensitivity and specificity of data at mixed thresholds were 95.33% (95% CI 90.60–100) and 92.01% (95% CI 87.61–96.42), respectively.

**Eye-level analysis.** A total of seven evaluations of AI for RDR screening were performed with data from seven studies and a total of 15 390 eyes. We only included the Aravind data from the Gulshan 2019 study because data from Sankara differed from our eligibility criteria.

The forest plot (Fig 6) shows moderate variation in the estimates of sensitivity and minimal variation in specificity. The HSROC plot (Fig 7) reveals good test accuracy since most study points lie in the upper left corner of the plot. Meta-analytical sensitivity and specificity of data at mixed thresholds were 91.24% (95% CI 79.15–100) and 93.90% (95% CI 90.63–97.16), respectively.

**Exploring heterogeneity.** We performed subgroup analyses to explore potential sources of heterogeneity only on the main analysis (patient level), consisting of ten studies, since the

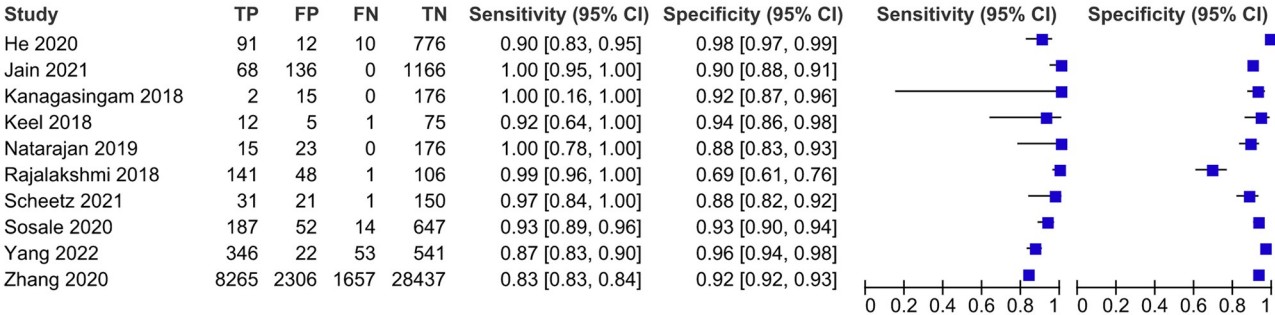

**Fig 4. Coupled forest plot of included studies for patient-level analysis.**

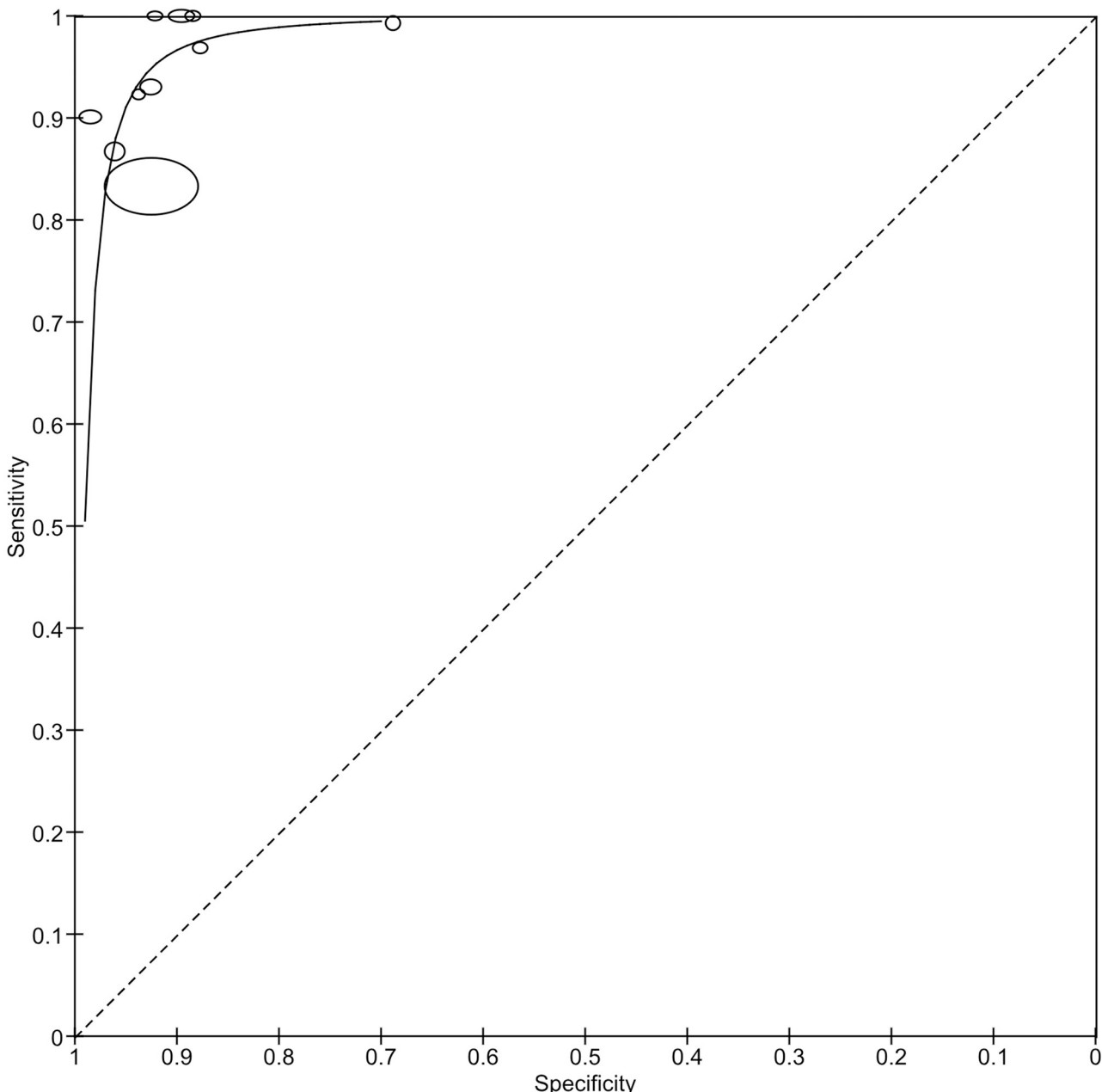

**Fig 5. HSROC plot of sensitivity vs specificity of AI for detecting RDR on patient-level analysis.**

data for the subgroups were more complete. A detailed result of subgroup analyses investigating potential sources of study-level heterogeneity is shown in Table 3.

*Level of economic development.* We classified the level of economic development of the countries included in our study using classification by the World Bank Group [36]. Of the ten studies included, three were conducted in HICs, and seven were conducted in LMICs. Australia was classified as a high-income country (HIC), and China and India, as lower- and middle-income country (LMIC). The sensitivity and specificity of AI in the real-world screening for

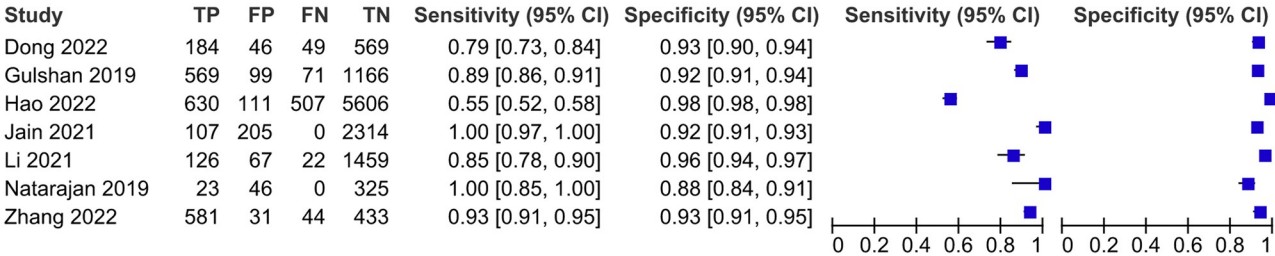

| Study | TP | FP | FN | TN | Sensitivity (95% CI) | Specificity (95% CI) |
|---|---|---|---|---|---|---|
| Dong 2022 | 184 | 46 | 49 | 569 | 0.79 [0.73, 0.84] | 0.93 [0.90, 0.94] |
| Gulshan 2019 | 569 | 99 | 71 | 1166 | 0.89 [0.86, 0.91] | 0.92 [0.91, 0.94] |
| Hao 2022 | 630 | 111 | 507 | 5606 | 0.55 [0.52, 0.58] | 0.98 [0.98, 0.98] |
| Jain 2021 | 107 | 205 | 0 | 2314 | 1.00 [0.97, 1.00] | 0.92 [0.91, 0.93] |
| Li 2021 | 126 | 67 | 22 | 1459 | 0.85 [0.78, 0.90] | 0.96 [0.94, 0.97] |
| Natarajan 2019 | 23 | 46 | 0 | 325 | 1.00 [0.85, 1.00] | 0.88 [0.84, 0.91] |
| Zhang 2022 | 581 | 31 | 44 | 433 | 0.93 [0.91, 0.95] | 0.93 [0.91, 0.95] |

**Fig 6. Coupled forest plot of included studies for eye-level analysis.**

**Fig 7. HSROC plot of sensitivity vs specificity of AI for detecting RDR on eye-level analysis.**

**Table 3. Subgroup analyses for the accuracy of AI in detecting RDR compared with trained HGs on patient-level analysis.**

| Analysis | | № of Studies | № of Participants | Sensitivity (95% CI) | Specificity (95% CI) |
|---|---|---|---|---|---|
| **Overall Meta-analysis** | | | | | |
| Patient-level | | 10 | 45 785 | 95.33% (90.60–100) | 92.01% (87.61–96.42) |
| **Subgroup Analyses** | | | | | |
| World Bank Country Classification | LMIC | 7 | 45 296 | 95.38% (90.38–100) | 92.21% (87.19–97.23) |
| | HIC | 3 | 489 | 95.61% (89.44–100) | 90.82% (87.76–93.87) |
| Level of Health- care Setting [a] | Primary | 4 | 2666 | 99.35% (96.85–100) | 93.72% (88.83–98.61) |
| | Tertiary | 4 | 2251 | 94.71% (89.00–100) | 90.88% (83.22–98.53) |
| DR Classification Criteria | ICDR | 8 | 45 489 | 95.44% (90.70–100) | 92.21% (87.80–96.62) |
| | NHS DES | 2 | 296 | 95.49% (89.19–100) | 89.85% (84.93–94.77) |

[a] No studies reported on secondary healthcare settings, thus not included in this table.

Data calculated using SAS Studio.

**CI**, Confidence Interval; **HG**, human grader; **HIC**, high-income country; **ICDR**, International Clinical Diabetic Retinopathy; **LMIC**, lower- and middle-income country; **NHS DES**, National Health Service Diabetic Eye Screening; **RDR**, referable diabetic retinopathy.

RDR in LMICs were 95.38% and 92.21%, respectively, and in HIC, they were 95.61% and 90.82%, respectively (Fig 8).

*Level of healthcare setting.* Four studies were done solely in tertiary-level healthcare settings, four in primary-level healthcare settings, and two at both levels (which were not included in this analysis). The sensitivity and specificity of AI in the real-world screening for RDR in

**LMIC**

| Study | TP | FP | FN | TN | Sensitivity (95% CI) | Specificity (95% CI) |
|---|---|---|---|---|---|---|
| He 2020 | 91 | 12 | 10 | 776 | 0.90 [0.83, 0.95] | 0.98 [0.97, 0.99] |
| Jain 2021 | 68 | 136 | 0 | 1166 | 1.00 [0.95, 1.00] | 0.90 [0.88, 0.91] |
| Natarajan 2019 | 15 | 23 | 0 | 176 | 1.00 [0.78, 1.00] | 0.88 [0.83, 0.93] |
| Rajalakshmi 2018 | 141 | 48 | 1 | 106 | 0.99 [0.96, 1.00] | 0.69 [0.61, 0.76] |
| Sosale 2020 | 187 | 52 | 14 | 647 | 0.93 [0.89, 0.96] | 0.93 [0.90, 0.94] |
| Yang 2022 | 346 | 22 | 53 | 541 | 0.87 [0.83, 0.90] | 0.96 [0.94, 0.98] |
| Zhang 2020 | 8265 | 2306 | 1657 | 28437 | 0.83 [0.83, 0.84] | 0.92 [0.92, 0.93] |

**HIC**

| Study | TP | FP | FN | TN | Sensitivity (95% CI) | Specificity (95% CI) |
|---|---|---|---|---|---|---|
| Kanagasingam 2018 | 2 | 15 | 0 | 176 | 1.00 [0.16, 1.00] | 0.92 [0.87, 0.96] |
| Keel 2018 | 12 | 5 | 1 | 75 | 0.92 [0.64, 1.00] | 0.94 [0.86, 0.98] |
| Scheetz 2021 | 31 | 21 | 1 | 150 | 0.97 [0.84, 1.00] | 0.88 [0.82, 0.92] |

**Fig 8. Coupled forest plots showing the subgroups in the level of economic development according to the World Bank country classification.** **HIC**, high-income country; **LMIC**, lower- and middle-income country.

**Primary Level**

| Study | TP | FP | FN | TN | Sensitivity (95% CI) | Specificity (95% CI) |
|---|---|---|---|---|---|---|
| He 2020 | 91 | 12 | 10 | 776 | 0.90 [0.83, 0.95] | 0.98 [0.97, 0.99] |
| Jain 2021 | 68 | 136 | 0 | 1166 | 1.00 [0.95, 1.00] | 0.90 [0.88, 0.91] |
| Kanagasingam 2018 | 2 | 15 | 0 | 176 | 1.00 [0.16, 1.00] | 0.92 [0.87, 0.96] |
| Natarajan 2019 | 15 | 23 | 0 | 176 | 1.00 [0.78, 1.00] | 0.88 [0.83, 0.93] |

**Tertiary Level**

| Study | TP | FP | FN | TN | Sensitivity (95% CI) | Specificity (95% CI) |
|---|---|---|---|---|---|---|
| Keel 2018 | 12 | 5 | 1 | 75 | 0.92 [0.64, 1.00] | 0.94 [0.86, 0.98] |
| Rajalakshmi 2018 | 141 | 48 | 1 | 106 | 0.99 [0.96, 1.00] | 0.69 [0.61, 0.76] |
| Sosale 2020 | 187 | 52 | 14 | 647 | 0.93 [0.89, 0.96] | 0.93 [0.90, 0.94] |
| Yang 2022 | 346 | 22 | 53 | 541 | 0.87 [0.83, 0.90] | 0.96 [0.94, 0.98] |

**Fig 9. Coupled forest plots showing the subgroups in the level of healthcare settings.**

primary-level healthcare settings were slightly higher than in tertiary-level (99.35% vs 94.71%, and 93.72% vs 90.88%, respectively) (Fig 9).

*DR classification criteria.* Eight studies used ICDR or its equivalence as DR classification criteria, and only two used the NHS DES criteria. It is important to note that doing a subgroup in this covariate does not intend to compare the two criteria but rather to see the robustness of AI in screening for RDR, even using different criteria. The sensitivity and specificity of AI in the real-world screening for RDR using ICDR were 95.45% and 92.21%, respectively, and using NHS DES, they were 95.49% and 89.85%, respectively, which did not show any significant variation (Fig 10).

**Sensitivity analysis.** We performed sensitivity analyses on two conditions stated below. A detailed result of sensitivity analyses is shown in Table 4.

*Inclusion of DME on the RDR definition.* After excluding four studies that did not include DME as part of the RDR definition, pooled sensitivity and specificity did not show any significant variation compared to the overall main meta-analysis (95.51% vs 95.33% and 91.35% vs 92.01%, respectively) (Fig 11).

**ICDR**

| Study | TP | FP | FN | TN | Sensitivity (95% CI) | Specificity (95% CI) |
|---|---|---|---|---|---|---|
| He 2020 | 91 | 12 | 10 | 776 | 0.90 [0.83, 0.95] | 0.98 [0.97, 0.99] |
| Jain 2021 | 68 | 136 | 0 | 1166 | 1.00 [0.95, 1.00] | 0.90 [0.88, 0.91] |
| Kanagasingam 2018 | 2 | 15 | 0 | 176 | 1.00 [0.16, 1.00] | 0.92 [0.87, 0.96] |
| Natarajan 2019 | 15 | 23 | 0 | 176 | 1.00 [0.78, 1.00] | 0.88 [0.83, 0.93] |
| Rajalakshmi 2018 | 141 | 48 | 1 | 106 | 0.99 [0.96, 1.00] | 0.69 [0.61, 0.76] |
| Sosale 2020 | 187 | 52 | 14 | 647 | 0.93 [0.89, 0.96] | 0.93 [0.90, 0.94] |
| Yang 2022 | 346 | 22 | 53 | 541 | 0.87 [0.83, 0.90] | 0.96 [0.94, 0.98] |
| Zhang 2020 | 8265 | 2306 | 1657 | 28437 | 0.83 [0.83, 0.84] | 0.92 [0.92, 0.93] |

**NHS DES**

| Study | TP | FP | FN | TN | Sensitivity (95% CI) | Specificity (95% CI) |
|---|---|---|---|---|---|---|
| Keel 2018 | 12 | 5 | 1 | 75 | 0.92 [0.64, 1.00] | 0.94 [0.86, 0.98] |
| Scheetz 2021 | 31 | 21 | 1 | 150 | 0.97 [0.84, 1.00] | 0.88 [0.82, 0.92] |

**Fig 10. Coupled forest plots showing the subgroups in the DR classification criteria. ICDR**, International Clinical Diabetic Retinopathy; **NHS DES**, National Health Service Diabetic Eye Screening.

**Table 4. Sensitivity analyses for the accuracy of AI in detecting RDR compared with trained HGs on patient-level analysis.**

| Analysis | № of Studies | № of Participants | Sensitivity (95% CI) | Specificity (95% CI) |
|---|---|---|---|---|
| **Overall Meta-analysis** | | | | |
| Patient-level | 10 | 45 785 | 95.33% (90.60–100) | 92.01% (87.61–96.42) |
| **Sensitivity Analyses** | | | | |
| Only studies with DME included in the RDR definition | 6 | 2595 | 95.51% (92.58–98.44) | 91.35% (84.92–97.78) |
| Only studies with a total of ≥3 HGs | 8 | 45 378 | 94.69% (90.11–99.28) | 92.37% (87.93–96.81) |

Data calculated using SAS Studio.

**CI**, confidence interval; **DME**, diabetic macular edema; **HG**, human grader; **RDR**, referable diabetic retinopathy.

**DME Included**

| Study | TP | FP | FN | TN | Sensitivity (95% CI) | Specificity (95% CI) | Sensitivity (95% CI) | Specificity (95% CI) |
|---|---|---|---|---|---|---|---|---|
| He 2020 | 91 | 12 | 10 | 776 | 0.90 [0.83, 0.95] | 0.98 [0.97, 0.99] | | |
| Keel 2018 | 12 | 5 | 1 | 75 | 0.92 [0.64, 1.00] | 0.94 [0.86, 0.98] | | |
| Natarajan 2019 | 15 | 23 | 0 | 176 | 1.00 [0.78, 1.00] | 0.88 [0.83, 0.93] | | |
| Rajalakshmi 2018 | 141 | 48 | 1 | 106 | 0.99 [0.96, 1.00] | 0.69 [0.61, 0.76] | | |
| Scheetz 2021 | 31 | 21 | 1 | 150 | 0.97 [0.84, 1.00] | 0.88 [0.82, 0.92] | | |
| Sosale 2020 | 187 | 52 | 14 | 647 | 0.93 [0.89, 0.96] | 0.93 [0.90, 0.94] | | |

**Fig 11. Coupled forest plot of studies that include DME on the RDR definition. DME**, diabetic macular edema; **RDR**, referable diabetic retinopathy.

*Total number of human graders.* After excluding two studies that have a total of ≤2 HGs as the ground truth, pooled sensitivity and specificity also did not show any significant variation compared to the overall main meta-analysis (94.69% vs 95.33% and 92.37% vs 92.01%, respectively) (Fig 12).

**Investigation of publication bias.** We did not investigate publication bias since, according to Salameh, et al. [16], the statistical investigation of publication and reporting bias is not routinely recommended in systematic reviews involving DTA.

**Human Graders ≥ 3**

| Study | TP | FP | FN | TN | Sensitivity (95% CI) | Specificity (95% CI) | Sensitivity (95% CI) | Specificity (95% CI) |
|---|---|---|---|---|---|---|---|---|
| He 2020 | 91 | 12 | 10 | 776 | 0.90 [0.83, 0.95] | 0.98 [0.97, 0.99] | | |
| Jain 2021 | 68 | 136 | 0 | 1166 | 1.00 [0.95, 1.00] | 0.90 [0.88, 0.91] | | |
| Keel 2018 | 12 | 5 | 1 | 75 | 0.92 [0.64, 1.00] | 0.94 [0.86, 0.98] | | |
| Rajalakshmi 2018 | 141 | 48 | 1 | 106 | 0.99 [0.96, 1.00] | 0.69 [0.61, 0.76] | | |
| Scheetz 2021 | 31 | 21 | 1 | 150 | 0.97 [0.84, 1.00] | 0.88 [0.82, 0.92] | | |
| Sosale 2020 | 187 | 52 | 14 | 647 | 0.93 [0.89, 0.96] | 0.93 [0.90, 0.94] | | |
| Yang 2022 | 346 | 22 | 53 | 541 | 0.87 [0.83, 0.90] | 0.96 [0.94, 0.98] | | |
| Zhang 2020 | 8265 | 2306 | 1657 | 28437 | 0.83 [0.83, 0.84] | 0.92 [0.92, 0.93] | | |

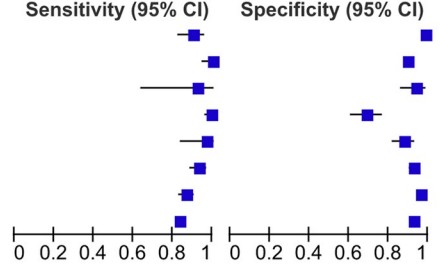

**Fig 12. Coupled forest plot of studies with ≥3 human graders as the ground truth on reference standard.**

## Discussion

### Summary of main findings

Artificial intelligence screening incorporating a range of software applications has been evaluated for detecting RDR in real-world settings. Studies in this review came from various economic settings and level of health care, all using recognised DR classification criteria. This review provides evidence, for the first time from prospective studies, for the effectiveness of AI in screening for RDR, in real-world settings.

This review aimed to assess the accuracy of AI solutions in detecting RDR in different resource settings. We found no variation in the diagnostic accuracy of AI, whether deployed in the LMICs or HICs, meaning AI in screening for RDR can be used universally. In our review, all AI models used in the primary studies, whether conducted in LMICs or HICs, utilised deep learning algorithms which were all pre-trained (except for two studies which did not provide details) and were all fine-tuned using datasets of sufficient quantity and quality. Also, most training datasets used were from Eye Picture Archive Communication System (EyePACS), a database specifically for DR screening and diagnosis, containing more than five million fundus photographs of diverse populations with various stages of DR, and requiring their photographers to be certified; thus ensuring good representation of quality images for pre-training and fine-tuning [37]. All these contribute to the high diagnostic accuracy of AI in detecting RDR in both LMICs and HICs.

Regarding different DR classifications used, we found no variation between ICDR and NHS DES because both the AI models and HGs used the same criteria when grading RDR. Thus, stakeholders need to note that when integrating an AI model into a DR screening programme, the DR criteria used to train the AI model should be the same as the DR criteria used by the trained HGs in that setting or country to prevent misclassifications.

However, on the level of the healthcare setting, studies done in the primary-level healthcare settings have higher diagnostic accuracy compared to those done in tertiary-level healthcare settings. One of the reasons may be having more patients with advanced disease or other comorbidities in tertiary care settings where using AI in screening for RDR can be more challenging. Another possible reason is that screening programmes for detecting RDR are mostly conducted in primary-level healthcare settings, where it is usually intended to be; thus, primary-level healthcare settings may have better structured protocols leading to more efficient and well-organised implementation in integrating AI for detecting RDR compared to tertiary-level healthcare settings.

We applied the summary estimates to a hypothetical cohort of 1000 patients to our main analysis using the Grading of Recommendations, Assessment, Development and Evaluation (GRADE)pro guideline development tool [38] (Table 5). Our findings suggest that if AI is used for the detection of RDR in real-world settings, 95% of patients with RDR will be correctly screened positive for the condition, and 92% of patients with no RDR will be correctly screened negative for the condition. We were interested in knowing the number of patients who will be correctly and unnecessarily referred to tertiary healthcare for a further eye examination. For our prevalence rate, we used a prevalence estimate of 6.5%, which is from a recent multi-ethnic study involving datasets from Singapore, the USA, China and Australia [39] and a prevalence estimate of 2%, which is the national prevalence estimate of RDR in India [40]. Using an RDR prevalence of 6.5%, AI will correctly detect RDR in 62 patients living with diabetes, miss detecting three RDR cases while unnecessarily refer 75 patients living with diabetes without RDR, for further examination.

We also explored the effect of excluding studies that did not include DME in the RDR definition and found that the diagnostic accuracy of AI has no significant variation; this does not

**Table 5. Summary of findings of the review evaluated using the GRADEpro GDT.**

**Review question**: What is the diagnostic test accuracy of AI in screening for RDR compared with trained HGs among patients with diabetes in real-world settings?

**Population**: People living with clinically diagnosed type 1 and type 2 diabetes

**Setting**: Real-world settings

**Index test**: Artificial intelligence

**Reference standard**: Trained HGs

**Study design**: Cross-sectional studies with prospective data collection

**Total № of studies**: 15 studies; **Patient-level (Main) analysis**: 10 studies (45 785 patients); **Eye-level analysis**: 7 studies (15 390 eyes)

| Effect (95% CI) | Test Result | № of results per 1000 Samples Tested (95% CI) | | № of Samples (Studies) | Certainty of the Evidence (GRADE) |
|---|---|---|---|---|---|
| | | Prevalence 2% [a] | Prevalence 6.5% [b] | | |
| **Patient-level analysis** | | | | | |
| **Pooled sensitivity** 95% (91–100%) | **True Positive** | **19** (18–20) | **62** (59–65) | 10 985 patients (10 studies) | ⊕⊕⊕◯ **MODERATE** [c] |
| | **False Negative** | **1** (0–2) | **3** (0–6) | | |
| **Pooled specificity** 92% (88–96%) | **True Negative** | **902** (859–945) | **860** (819–902) | 34 890 patients (10 studies) | ⊕⊕⊕◯ **MODERATE** [c] |
| | **False Positive** | **78** (35–121) | **75** (33–116) | | |
| **Eye-level analysis** | | | | | |
| **Pooled sensitivity** 91% (79–100%) | **True Positive** | **18** (16–20) | **59** (51–65) | 2913 eyes (7 studies) | ⊕◯◯◯ **VERY LOW** [d, e, f] |
| | **False Negative** | **2** (0–4) | **6** (0–14) | | |
| **Pooled specificity** 94% (91–97%) | **True Negative** | **920** (888–952) | **878** (847–908) | 12 477 eyes (7 studies) | ⊕⊕⊕◯ **MODERATE** [d] |
| | **False Positive** | **60** (28–92) | **57** (27–88) | | |

Prevalence data calculated using GRADEpro GDT.

[a] National prevalence estimate of RDR in India [40]

[b] Prevalence estimate of RDR in a multi-ethnic study involving datasets from Singapore, the USA, Hong Kong, China and Australia [39]

[c] **Risk of bias (-1)**: QUADAS-2 tool was used to assess for the risk of bias in the 10 studies. In the domain of *Patient Selection*, the risk of bias was high in 1 study and was unclear in 8; In the domain of *Index Test*, it was high in 4 studies and unclear in 6; In the domain of *Reference Standard*, it was high in 2 studies and unclear in 4; and in the domain of *Flow and Timing*, it was high in 1 study.

[d] **Risk of bias (-1)**: Risk of bias was assessed in the 7 studies of this level of analysis. In the domain of *Patient Selection*, the risk of bias was high in 2 studies and was unclear in 5; In the domain of *Index Test*, it was high in 2 studies and unclear in 4; In the domain of *Reference Standard*, it was high in 2 studies and unclear in 1; and in the domain of *Flow and Timing*, it was high in 1 study.

[e] **Inconsistency(-1)**: Statistical heterogeneity based on the forest plot showed moderate variation in the sensitivity.

[f] **Imprecision (-1)**: The CI of the pooled sensitivity is wide, indicating that there is an uncertainty in the estimate and that the true value could potentially be lower.

**Grade Definition** [38]

**High**: Further research is very unlikely to change our confidence in the estimate of effect; **Moderate**: Further research is likely to have an important impact on our confidence in the estimate of effect and may change the estimate;

**Low**: Further research is very likely to have an important impact on our confidence in the estimate of effect and is likely to change the estimate; **Very low**: Any estimate of effect is very uncertain.

**AI**, artificial intelligence; **CI**, Confidence Interval; **DME**, diabetic macular edema; **DR**, diabetic retinopathy; **GDT**, guideline development tool; **GRADE**, Grading of Recommendations, Assessment, Development and Evaluation; **HG**, human grader; **RDR**, referable diabetic retinopathy

mean that inclusion or exclusion of DME is nonsignificant in screening for RDR, rather, this is because trained HGs adhered to the grading protocol of the study with regards to RDR definition. Furthermore, the effect of excluding studies with ≤2 trained HGs (ophthalmologists and trained and certified HGs from retina reading centres) did not affect the diagnostic accuracy of the data. This review, thus, highlights the importance of trained HGs acting as a reference standard for grading the fundus images.

### Strengths and limitations of this review

**Strengths.** This is the first systematic review and meta-analysis assessing the diagnostic accuracy of AI in screening for RDR in real-world settings that included studies using prospective data collection. We did not restrict our literature search in terms of language and publication year to minimise the chance of missing studies. We were able to present the accuracy estimates in patient-level and eye-level analysis, rather than just combining these data to prevent unit-of-analysis issues and avoid bias in precision. We were able to tailor and pilot our QUADAS-2 tool to our study, adding more signalling questions to fit AI studies since QUADAS-AI by Sounderajah et al. [41] was not yet published during the time of our review. Data extraction and assessment of the risk of bias were performed by two review authors, thus, reducing the risk of bias. We avoided all case-control studies since studies involving a control group without RDR and patients with RDR may exaggerate the diagnosis accuracy [17]. We included studies using different DR criteria (rather than just restricting to certain criteria), where results showed no significant variation in the accuracy estimates.

**Limitations.** *Eligibility*. Our definition of RDR in this review is according to how the authors of the primary studies defined them, with or without DME. In the clinical setting, cases of DME should be referred for further examination when detected. However, to accurately detect DME, optical coherence tomography (OCT), which gives detailed 3D images of the eye, is the gold standard; thus, this makes fundus images less advantageous as it only provides 2D images. Therefore, it is important that further studies be done for AI models to be trained and developed to read OCT together with fundus images for higher accuracy and better applicability.

*Quality of included studies*. All eligible studies had either an unclear risk or, a high risk of bias in at least one of the QUADAS-2 domains. Amongst the included studies, 80% did not report as to how patients were enrolled in the study, making them unclear. Also, many of the studies did not clearly report a pre-specified threshold which may influence the diagnostic accuracy of the test if the authors select a positivity cut-off after obtaining the results. Thus, we support that DTA studies following the Standards for Reporting of Diagnostic Accuracy Studies (STARD) guidelines by Cohen et al. [42] or the proposed STARD-AI guidelines by Sounderajah et al. [43], when available, to avoid these uncertainties.

Regarding the QUADAS-2 domain on flow and timing, specifically as regards the signalling question relating to whether all enrolled patients were included in the analysis, we deemed a study as high risk if the discrepancies between the enrolled and analysed patients were not motivated, or were related to the severity of RDR (even though most studies have excluded ungradable images from the analysis). This was done since including ungradable images may lead to inaccuracy and not give meaningful results. Therefore, it is important that during the implementation of AI in DR screening programmes, the protocol for evaluating images as ungradable should be available, (e.g. considering mydriasis, if needed, assuring quality images when capturing photos, etc.), to avoid missed detections and unnecessary referrals since during DR screening, patients with fundus images deemed ungradable by AI should also be referred to ophthalmologists for proper assessment.

Another limitation found is the representativity of the level of economic development by World Bank country classification. The subgroup for HIC is represented only by Australia, and the subgroup LMIC, only by China and India. Although there were DTA studies conducted in other countries (i.e. USA, Spain, Zambia, etc), they were, unfortunately, excluded against our eligibility criteria.

### Applicability of findings to the review question

Concerns regarding the applicability of all included studies were deemed low, except for two studies that were not able to avoid inappropriate exclusions. We assessed the applicability of findings to our review question with low concerns since all studies included AI models that were able to detect RDR in real-world settings; included patients were all clinically diagnosed with type 1 and/or type 2 diabetes; the grading of the same images was all compared to the grading of the trained HGs.

## Conclusion

Our review provides evidence that AI could effectively screen for RDR even in real-world settings. Whether in the HICs or LMICs, the detection of RDR using AI in real-world settings is highly sensitive and specific. It has higher accuracy when deployed at the primary-level than in tertiary-level healthcare settings.

## Implications for practice

Although AI in screening for DR has been showing promising results, it is important to consider where to deploy them. Patient-wise, it will be able to screen more patients living with diabetes, leading to early diagnosis and treatment. It can also increase disease awareness, promoting a healthy lifestyle and diabetes control to these patients. However, healthcare-wise, AI might be unnecessarily referring a handful of patients without RDR to tertiary healthcare centres. In HICs, where manpower is usually not an issue, this might not be a problem; however, in LMICs, where it is a challenge, referring false positive cases to the already few and straining eye health workers can overburden them. Thus, we recommend a clinical pathway in these low-resource settings, where trained or certified lay graders in primary healthcare can countercheck all the fundus images of patients who screened positive for RDR before officially referring them, rather than just leaving the referral decisions to the AI system.

## Implications for research

In recent years, researchers and clinicians have been advocating the use of real-world performance of AI for healthcare to evaluate further their real impact on image quality and system usability rather than just validating them using retrospective high-quality databases [29]. Our review was able to pool the diagnostic accuracy of AI in screening RDR of studies using prospective data collection; therefore, can provide recommendations to evidence-based guidelines to integrate AI in DR screening programmes in real-world settings. We recommend further studies on integrating OCT aside from using fundus imaging in AI algorithms so screening for DME will be more accurate.

## Supporting information

**S1 Table. Search strategy.**
(DOCX)

**S1 Checklist. PRISMA-DTA checklist.**
(DOCX)

## Author Contributions

**Conceptualization:** Holijah Uy, Abraham Opare, Deon Minnies.

**Data curation:** Holijah Uy, Christopher Fielding, Ameer Hohlfeld, Eleanor Ochodo, Elton Mukonda, Mark E. Engel.

**Formal analysis:** Holijah Uy, Eleanor Ochodo.

**Methodology:** Holijah Uy, Christopher Fielding, Ameer Hohlfeld, Eleanor Ochodo, Deon Minnies, Mark E. Engel.

**Supervision:** Ameer Hohlfeld, Eleanor Ochodo, Elton Mukonda, Deon Minnies, Mark E. Engel.

**Validation:** Christopher Fielding, Ameer Hohlfeld, Eleanor Ochodo, Deon Minnies, Mark E. Engel.

**Visualization:** Mark E. Engel.

**Writing – original draft:** Holijah Uy.

**Writing – review & editing:** Holijah Uy, Christopher Fielding, Ameer Hohlfeld, Eleanor Ochodo, Abraham Opare, Elton Mukonda, Deon Minnies, Mark E. Engel.

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
