## [Decision Letter · Decision Letter 0]

17 Jul 2023

PGPH-D-23-01159

Diagnostic test accuracy of artificial intelligence in screening for referable diabetic retinopathy in real-world settings: A systematic review and meta-analysis

Dear Dr. Uy,

Thank you for submitting your manuscript to PLOS Global Public Health. After careful consideration, we feel that it has merit but does not fully meet PLOS Global Public Health’s publication criteria as it currently stands. Therefore, we invite you to submit a revised version of the manuscript that addresses the points raised during the review process.

We look forward to receiving your revised manuscript.

Kind regards,

Segun Fatumo, PhD

Academic Editor

Journal Requirements:

Additional Editor Comments (if provided):

Reviewers' comments:

Reviewer's Responses to Questions

**Comments to the Author**

1. Does this manuscript meet PLOS Global Public Health’s publication criteria? Is the manuscript technically sound, and do the data support the conclusions? The manuscript must describe methodologically and ethically rigorous research with conclusions that are appropriately drawn based on the data presented.

Reviewer #1: Yes

Reviewer #2: Yes

2. Has the statistical analysis been performed appropriately and rigorously?

Reviewer #1: Yes

Reviewer #2: Yes

3. Have the authors made all data underlying the findings in their manuscript fully available (please refer to the Data Availability Statement at the start of the manuscript PDF file)?

Reviewer #1: Yes

Reviewer #2: Yes

4. Is the manuscript presented in an intelligible fashion and written in standard English?

Reviewer #1: Yes

Reviewer #2: Yes

5. Review Comments to the Author

Reviewer #1: Thank you for your submission.

The topic is quite relevant and will help form further guidelines.

The paper is very well drafted and follows the established protocol.

I would recommend this paper for publishing and to be highlighted in media and journal website.

Reviewer #2: Abstract

The abstract is well-written and provides information about the study's methods, results, and interpretation. It presents the aim, methodology, key findings, and conclusions succinctly. However, here are a few suggestions to improve the clarity and readability:

- Be more specific with terms like 'moderate increase' and 'minimal decrease'. If possible, provide exact numbers or percentages to give the reader a clear understanding of the changes.

- The conclusion might benefit from more explicit statements about the implications of the findings. You've mentioned that AI is effective in screening for RDR, but you could discuss how this impacts current practice or what the next steps could be.

- Line 35: The abbreviation SROC, did you want to write HSROC?

Introduction

The introduction is detailed and provides a good overview of the topic, clearly establishing the importance and urgency of the research. The problem is well-defined, and the literature gap that this study intends to address is well-articulated. However, a few areas could be improved:

- A brief historical context could help to put this research’s importance into perspective. For instance, what has been the trajectory of DR treatments? What has been the impact of AI in healthcare generally?

- Line 80-81: Ensure all the information presented has appropriate and recent references. For example, the statement "In recent years, retrospective validation studies have shown AI to have high diagnostic accuracy in detecting DR..." needs a reference citation.

Methods

The Methods section of the systematic review appears to be largely comprehensive and transparent in its description of the processes and criteria used for the review. It carefully outlines the databases and search strategies, eligibility criteria, selection of studies, data extraction, and risk assessment methods, all adhering to the Preferred Reporting Items for Systematic Review and Meta-analysis of Diagnostic Test Accuracy Studies (PRISMA-DTA) guidelines. This is commendable as it enhances the transparency and replicability of the systematic review.

- Line 141-142; The title "Report characteristics" may not perfectly fit the content described in the paragraph. The text discusses specific criteria used to include or exclude studies for the systematic review, specifically the publication year and language, as well as the exclusion of study protocols. A section called ‘types of studies’ already discusses the studies to be included or excluded. The date range could be added to the databases searched and search strategies.

- Line 160-161 – please check the heading ‘risk of bias and acceptability’ first sentence that followed ‘risk of bias and applicability’. Can the authors confirm if acceptability and applicability are used interchangeably here?

- The authors should provide more clarity on resolving disagreements. While it's mentioned that a third reviewer would be consulted, the specific method or process (e.g., discussion, voting, senior author's decision) should be described.

- The authors write, ‘we also hand-searched the reference lists of relevant primary studies, systematic reviews, and the following journals: British Journal of Ophthalmology, American Journal of Ophthalmology, Ophthalmology and Retina, JAMA Ophthalmology, and Investigative Ophthalmology and Visual Science’. This search and the number of studies retrieved are not captured in the PRISMA flow chart or the findings. Can the authors clarify this?

- While visual inspection of forest plots and HSROC plots are mentioned, the authors could have considered adding a quantitative assessment of heterogeneity using I-squared (I²) statistics or Cochran's Q test might provide more accurate estimates of heterogeneity. Was there a reason this wasn’t done?

Discussion and conclusion

The Discussion section does an excellent job of summarizing the main findings and their implications and provides context for these findings. The authors present their findings clearly, discussing the overall effectiveness of AI in screening for Referable Diabetic Retinopathy (RDR) in real-world settings. They also engage in hypothetical cohort modelling, which is a positive aspect, giving the reader a more intuitive understanding of the study's results.

- The authors pointed out that their review provides evidence from prospective studies for the first time. They could elaborate on why this is significant and how their work contributes to or changes our current understanding of the field.

- The authors should provide more interpretation and discussion of their findings. For instance, why do they think the level of healthcare setting influenced diagnostic accuracy? Could factors like the complexity of cases, patient comorbidities, or healthcare workers' skill/experience level influence this result?

6. PLOS authors have the option to publish the peer review history of their article (what does this mean?). If published, this will include your full peer review and any attached files.

**Do you want your identity to be public for this peer review?** For information about this choice, including consent withdrawal, please see our Privacy Policy.

Reviewer #1: **Yes: **Manish Barman

Reviewer #2: **Yes: **Isaac Amankwaa

---

## [Decision Letter · Decision Letter 1]

24 Aug 2023

Diagnostic test accuracy of artificial intelligence in screening for referable diabetic retinopathy in real-world settings: A systematic review and meta-analysis

PGPH-D-23-01159R1

Dear Dr Uy,

We are pleased to inform you that your manuscript 'Diagnostic test accuracy of artificial intelligence in screening for referable diabetic retinopathy in real-world settings: A systematic review and meta-analysis' has been provisionally accepted for publication in PLOS Global Public Health.

Best regards,

Segun Fatumo, PhD

Academic Editor

Reviewer Comments (if any, and for reference):

Reviewer's Responses to Questions

**Comments to the Author**

1. If the authors have adequately addressed your comments raised in a previous round of review and you feel that this manuscript is now acceptable for publication, you may indicate that here to bypass the “Comments to the Author” section, enter your conflict of interest statement in the “Confidential to Editor” section, and submit your "Accept" recommendation.

Reviewer #2: All comments have been addressed

2. Does this manuscript meet PLOS Global Public Health’s publication criteria? Is the manuscript technically sound, and do the data support the conclusions? The manuscript must describe methodologically and ethically rigorous research with conclusions that are appropriately drawn based on the data presented.

Reviewer #2: Yes

3. Has the statistical analysis been performed appropriately and rigorously?

Reviewer #2: Yes

4. Have the authors made all data underlying the findings in their manuscript fully available (please refer to the Data Availability Statement at the start of the manuscript PDF file)?

Reviewer #2: Yes

5. Is the manuscript presented in an intelligible fashion and written in standard English?

Reviewer #2: Yes

6. Review Comments to the Author

Reviewer #2: Dear Authors,

I would like to extend my sincere gratitude for your diligent response to the key issues raised in my initial review. It is evident that you have made thoughtful corrections where necessary and provided valuable clarifications in areas that required further explanation. I must congratulate you on the great work you have done. Your effort in addressing the concerns has enhanced the overall quality and integrity of the manuscript. Once again, thank you for your hard work and cooperation in this review process.

Best regards.

7. PLOS authors have the option to publish the peer review history of their article (what does this mean?). If published, this will include your full peer review and any attached files.

**Do you want your identity to be public for this peer review?** For information about this choice, including consent withdrawal, please see our Privacy Policy.

Reviewer #2: **Yes: **Isaac Amankwaa (PhD, RN)
